# Presynaptic stochasticity improves energy efficiency and helps alleviate the stability-plasticity dilemma

**Simon Schug[1†]\*, Frederik Benzing[2†], Angelika Steger[2]**

[1]Institute of Neuroinformatics, University of Zurich & ETH Zurich, Zurich, Switzerland; [2]Department of Computer Science, ETH Zurich, Zurich, Switzerland

**Abstract** When an action potential arrives at a synapse there is a large probability that no neurotransmitter is released. Surprisingly, simple computational models suggest that these synaptic failures enable information processing at lower metabolic costs. However, these models only consider information transmission at single synapses ignoring the remainder of the neural network as well as its overall computational goal. Here, we investigate how synaptic failures affect the energy efficiency of models of entire neural networks that solve a goal-driven task. We find that presynaptic stochasticity and plasticity improve energy efficiency and show that the network allocates most energy to a sparse subset of important synapses. We demonstrate that stabilising these synapses helps to alleviate the stability-plasticity dilemma, thus connecting a presynaptic notion of importance to a computational role in lifelong learning. Overall, our findings present a set of hypotheses for how presynaptic plasticity and stochasticity contribute to sparsity, energy efficiency and improved trade-offs in the stability-plasticity dilemma.

**\*For correspondence:**
sschug@ethz.ch

[†]These authors contributed equally to this work

**Competing interests:** The authors declare that no competing interests exist.

## Introduction

It has long been known that synaptic signal transmission is stochastic (*del Castillo and Katz, 1954*). When an action potential arrives at the presynapse, there is a high probability that no neurotransmitter is released – a phenomenon observed across species and brain regions (*Branco and Staras, 2009*). From a computational perspective, synaptic stochasticity seems to place unnecessary burdens on information processing. Large amounts of noise hinder reliable and efficient computation (*Shannon, 1948*; *Faisal et al., 2005*) and synaptic failures appear to contradict the fundamental evolutionary principle of energy-efficient processing (*Niven and Laughlin, 2008*). The brain, and specifically action potential propagation consume a disproportionately large fraction of energy (*Attwell and Laughlin, 2001*; *Harris et al., 2012*) – so why propagate action potentials all the way to the synapse only to ignore the incoming signal there?

To answer this neurocomputational enigma various theories have been put forward, see *Llera-Montero et al., 2019* for a review. One important line of work proposes that individual synapses do not merely maximise information transmission, but rather take into account metabolic costs, maximising the information transmitted *per unit of energy* (*Levy and Baxter, 1996*). This approach has proven fruitful to explain synaptic failures (*Levy and Baxter, 2002*; *Harris et al., 2012*), low average firing rates (*Levy and Baxter, 1996*) as well as excitation-inhibition balance (*Sengupta et al., 2013*) and is supported by fascinating experimental evidence suggesting that both presynaptic glutamate release (*Savtchenko et al., 2013*) and postsynaptic channel properties (*Harris et al., 2015*; *Harris et al., 2019*) are tuned to maximise information transmission per energy.

However, so far information-theoretic approaches have been limited to signal transmission at single synapses, ignoring the context and goals in which the larger network operates. As soon as context and goals guide network computation certain pieces of information become more relevant than

others. For instance, when reading a news article the textual information is more important than the colourful ad blinking next to it – even when the latter contains more information in a purely information-theoretic sense.

Here, we study presynaptic stochasticity on the network level rather than on the level of single synapses. We investigate its effect on (1) energy efficiency and (2) the stability-plasticity dilemma in model neural networks that learn to selectively extract information from complex inputs.

We find that presynaptic stochasticity in combination with presynaptic plasticity allows networks to extract information at lower metabolic cost by sparsely allocating energy to synapses that are important for processing the given stimulus. As a result, presynaptic release probabilities encode synaptic importance. We show that this notion of importance is related to the Fisher information, a theoretical measure for the network's sensitivity to synaptic changes.

Building on this finding and previous work (*Kirkpatrick et al., 2017*), we explore a potential role of presynaptic stochasticity in the stability-plasticity dilemma. In line with experimental evidence (*Yang et al., 2009*; *Hayashi-Takagi et al., 2015*), we demonstrate that selectively stabilising important synapses improves lifelong learning. Furthermore, these experiments link presynaptically induced sparsity to improved memory.

## Model

Our goal is to understand how information processing and energy consumption are affected by stochasticity in synaptic signal transmission. While there are various sources of stochasticity in synapses, here, we focus on modelling *synaptic failures* where action potentials at the presynapse fail to trigger any postsynaptic depolarisation. The probability of such failures is substantial (*Branco and Staras, 2009*; *Hardingham et al., 2010*; *Sakamoto et al., 2018*) and, arguably, due to its all-or-nothing-characteristic has the largest effect on both energy consumption and information transmission.

As a growing body of literature suggests, artificial neural networks (ANNs) match several aspects of biological neuronal networks in various goal-driven situations (*Kriegeskorte, 2015*; *Yamins and DiCarlo, 2016*; *Kell et al., 2018*; *Banino et al., 2018*; *Cueva and Wei, 2018*; *Mattar and Daw, 2018*). Crucially, they are the only known model to solve complex vision and reinforcement learning tasks comparably well as humans. We therefore choose to extend this class of models by explicitly incorporating synaptic failures and study their properties in a number of complex visual tasks.

## Model details

The basic building blocks of ANNs are neurons that combine their inputs $a_1, \ldots, a_n$ through a weighted sum $w_1 a_1 + \ldots w_n a_n$ and apply a nonlinear activation function $\sigma(\cdot)$. The weights $w_i$ naturally correspond to *synaptic strengths* between presynaptic neuron $i$ and the postsynaptic neuron. Although synaptic transmission is classically described as a binomial process (*del Castillo and Katz, 1954*) most previous modelling studies assume the synaptic strengths to be deterministic. This neglects a key characteristic of synaptic transmission: the possibility of synaptic failures where no communication between pre- and postsynapse occurs at all.

In the present study, we explicitly model presynaptic stochasticity by introducing a random variable $r_i \sim \mathrm{Bernoulli}(p_i)$, whose outcome corresponds to whether or not neurotransmitter is released. Formally, each synapse $w_i$ is activated stochastically according to

$$w_i = \underbrace{r_i}_{\substack{\text{stochastic} \\ \text{release}}} \cdot \underbrace{m_i}_{\substack{\text{synaptic} \\ \text{strength}}}, \text{ where } r_i \sim \mathrm{Bernoulli} \underbrace{(p_i)}_{\substack{\text{release} \\ \text{probability}}} \tag{1}$$

so that it has expected synaptic strength $\bar{w}_i = p_i m_i$. The postsynaptic neuron calculates a stochastic weighted sum of its inputs with a nonlinear activation

$$\underbrace{a^{\mathrm{post}}}_{\substack{\text{postsynaptic} \\ \text{activation}}} = \sigma \Big( \sum_{i=1}^{n} w_i \underbrace{a_i^{\mathrm{pre}}}_{\substack{\text{i-th presynaptic} \\ \text{input}}} \Big). \tag{2}$$

During learning, synapses are updated and both synaptic strength and release probability are changed. We resort to standard learning rules to change the expected synaptic strength. For the

multilayer perceptron, this update is based on stochastic gradient descent with respect to a loss function $L(\bar{w}, p)$, which in our case is the standard cross-entropy loss. Concretely, we have

$$\bar{w}_i^{(t+1)} = \bar{w}_i^{(t)} - \eta g_i, \quad \text{where} \quad g_i = \frac{\partial L(\bar{w}^{(t)}, p)}{\partial \bar{w}_i^{(t)}} \tag{3}$$

where the superscript corresponds to time steps. Note that this update is applied to the expected synaptic strength $\bar{w}_i$, requiring communication between pre- and postsynape, see also Discussion. For the explicit update rule of the synaptic strength $m_i$ see Materials and methods, *Equation (8)*. For the standard perceptron model, $g_i$ is given by its standard learning rule (*Rosenblatt, 1958*). Based on the intuition that synapses which receive larger updates are more important for solving a given task, we update $p_i$ using the update direction $g_i$ according to the following simple scheme

$$p_i^{(t+1)} = \begin{cases} p_i^{(t)} + p_{\text{up}}, & \text{if } |g_i| > g_{\text{lim}}, \\ p_i^{(t)} - p_{\text{down}}, & \text{if } |g_i| \leq g_{\text{lim}}, \end{cases} \tag{4}$$

Here, $p_{\text{up}}, p_{\text{down}}, g_{\text{lim}}$ are three metaplasticity parameters shared between all synapses. (We point out that in a noisy learning setting the gradient $g$ does not decay to, so that the learning rule in (4) will maintain network function by keeping certain release probabilities high. See also Materials and methods for a theoretical analysis.) To prevent overfitting and to test robustness, we tune them using one learning scenario and keep them fixed for all other scenarios, see Materials and methods. To avoid inactivated synapses with release probability $p_i = 0$, we clamp $p_i$ to stay above 0.25, which we also use as the initial value of $p_i$ before learning.

On top of the above intuitive motivation, we give a theoretical justification for this learning rule in Materials and methods, showing that synapses with larger Fisher information obtain high release probabilities, also see Figure 2d.

## Measuring energy consumption

For our experiments, we would like to quantify the energy consumption of the neural network. *Harris et al., 2012* find that the main constituent of neural energy demand is synaptic signal transmission and that the cost of synaptic signal transmission is dominated by the energy needed to reverse postsynaptic ion fluxes. In our model, the component most closely matching the size of the postsynaptic current is the expected synaptic strength, which we therefore take as measure for the model's energy consumption. In the Appendix, we also measure the metabolic cost incurred by the activity of neurons by calculating their average rate of activity.

## Measuring information transmission

We would like to measure how well the neural network transmits information relevant to its behavioural goal. In particular, we are interested in the setting where the complexity of the stimulus is high relative to the amount of information that is relevant for the behavioural goal. To this end, we present complex visual inputs with high information content to the network and teach it to recognise the object present in the image. We then measure the mutual information between network output and object identity, see *Box 1*.

## Results

### Presynaptic stochasticity enables energy-efficient information processing

We now investigate the energy efficiency of a network that learns to classify digits from the MNIST handwritten digit dataset (*LeCun, 1998*). The inputs are high-dimensional with high entropy, but the relevant information is simply the identity of the digit. We compare the model with plastic, stochastic release to two controls. A standard ANN with deterministic synapses is included to investigate the combined effect of presynaptic stochasticity and plasticity. In addition, to isolate the effect of presynaptic plasticity, we introduce a control which has stochastic release, but with a fixed probability. In this control, the release probability is identical across synapses and chosen to match the average release probability of the model with plastic release after it has learned the task.

## Box 1. Mutual Information.

The Mutual Information $I(Y;Z)$ of two jointly distributed random variables $Y, Z$ is a common measure of their dependence (**Shannon, 1948**). Intuitively, mutual information captures how much information about $Y$ can be obtained from $Z$, or vice versa. Formally, it is defined as

$$I(Y;Z) \equiv H(Y) - H(Y|Z) = H(Z) - H(Z|Y)$$

Where $H(Y)$ is the entropy of $Y$ and $H(Y|Z)$ is the conditional entropy of $Y$ given $Z$. In our case, we want to measure how much task-relevant information $Y$ is contained in the neural network output $Z$. For example, the neural network might receive as input a picture of a digit with the goal of predicting the identity of the digit. Both the ground-truth digit identity $Y$ and the network's prediction $Z$ are random variables depending on the random image $X$. The measure $I(Y;Z)$ quantifies how much of the behaviourally relevant information $Y$ is contained in the network's prediction $Z$ ignoring irrelevant information also present in the complex, high-entropy image $X$.

All models are encouraged to find low-energy solutions by penalising large synaptic weights through standard $\ell_2$-regularisation. *Figure 1a* shows that different magnitudes of $\ell_2$-regularisation induce different information-energy trade-offs for all models, and that the model with plastic, stochastic release finds considerably more energy-efficient solutions than both controls, while the model with non-plastic release requires more energy then the deterministic model. Together, this supports the view that a combination of presynaptic stochasticity and plasticity promotes energy-efficient information extraction.

In addition, we investigate how stochastic release helps the network to lower metabolic costs. Intuitively, a natural way to save energy is to assign high release probabilities to synapses that are important to extract relevant information and to keep remaining synapses at a low release probability. *Figure 2a* shows that after learning, there are indeed few synapses with high release probabilities, while most release probabilities are kept low. We confirm that this sparsity develops

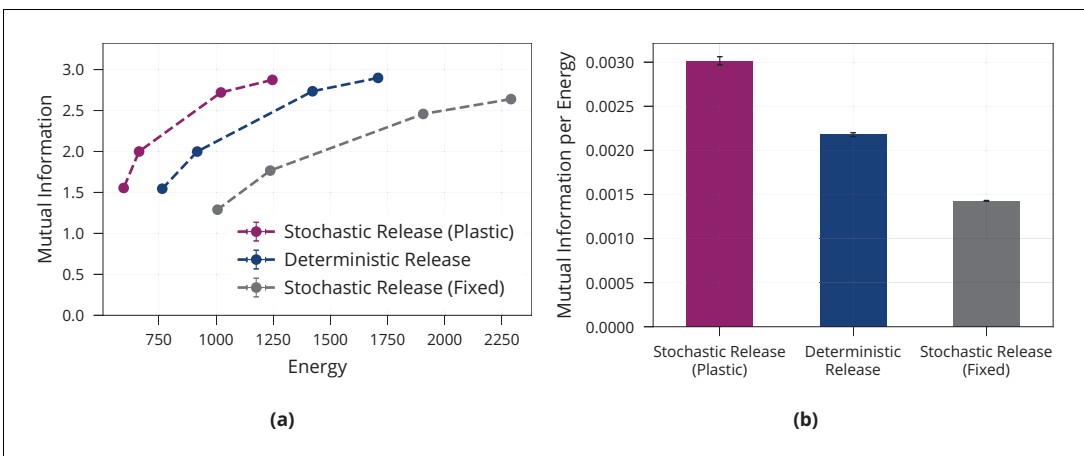

(a)   (b)

**Figure 1.** Energy efficiency of model with stochastic and plastic release. (a) Different trade-offs between mutual information and energy are achievable in all network models. Generally, stochastic synapses with learned release probabilities are more energy-efficient than deterministic synapses or stochastic synapses with fixed release probability. The fixed release probabilities model was chosen to have the same average release probability as the model with learned probabilities. (b) Best achievable ratio of information per energy for the three models from (a). Error bars in (a) and (b) denote the standard error for three repetitions of the experiment.

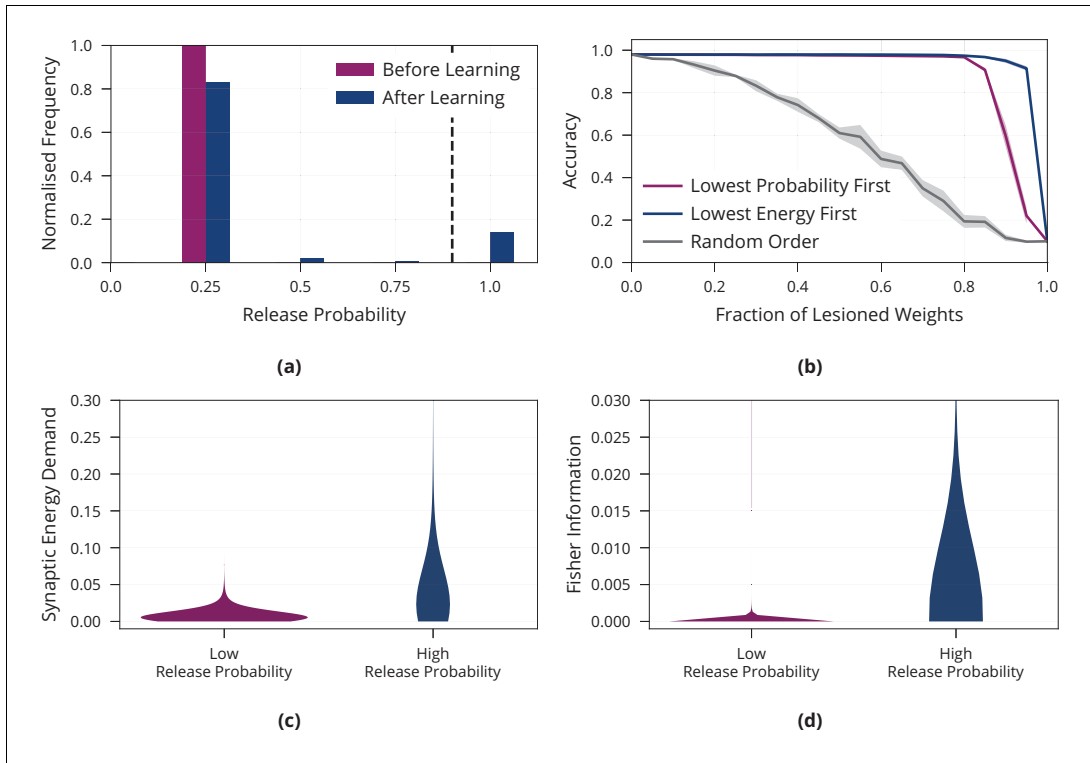

**Figure 2.** Importance of synapses with high release probability for network function. (**a**) Histogram of release probabilities before and after learning, showing that the network relies on a sparse subset of synapses to find an energy-efficient solution. Dashed line at $p = 0.9$ indicates our boundary for defining a release probability as 'low' or 'high'. We confirmed that results are independent of initial value of release probabilities before learning (see *Appendix 1—figure 2d*). (**b**) Accuracy after performing the lesion experiment either removing synapses with low release probabilities first or removing weights randomly, suggesting that synapses with high release probability are most important for solving the task. (**c**) Distribution of synaptic energy demand for high and low release probability synapses. (**d**) Distribution of the Fisher information for high and low release probability synapses. It confirms the theoretical prediction that high release probability corresponds to high Fisher information. All panels show accumulated data for three repetitions of the experiment. Shaded regions in (**b**) show standard error.

independently of the initial value of release probabilities before learning, see *Appendix 1—figure 1d*. To test whether the synapses with high release probabilities are most relevant for solving the task we perform a lesion experiment. We successively remove synapses with low release probability and measure how well the lesioned network still solves the given task, see *Figure 2b*. As a control, we remove synapses in a random order independent of their release probability. We find that maintaining synapses with high release probabilities is significantly more important to network function than maintaining random ones. Moreover, we find, as expected, that synapses with high release probabilities consume considerably more energy than synapses with low release probability, see *Figure 2c*. This supports the hypothesis that the model identifies important synapses for the task at hand and spends more energy on these synapses while saving energy on irrelevant ones.

We have seen that the network relies on a sparse subset of synapses to solve the task efficiently. However, sparsity is usually thought of on a neuronal level, with few neurons rather than few synapses encoding a given stimulus. Therefore, we quantify sparsity of our model on a neuronal level. For each neuron, we count the number of 'important' input- and output synapses, where we define 'important' to correspond to a release probability of at least $p = 0.9$. Note that the findings are robust with respect to the precise value of $p$, see *Figure 2a*. We find that the distribution of important synapses per neuron is inhomogeneous and significantly different from a randomly shuffled baseline with a uniform distribution of active synapses (Kolmogorov-Smirnoff test, $D = 0.505, p < 0.01$), see *Figure 3a*. Thus, some neurons have disproportionately many important inputs, while others have very few, suggesting sparsity on a neuronal level. As additional quantification of this effect, we

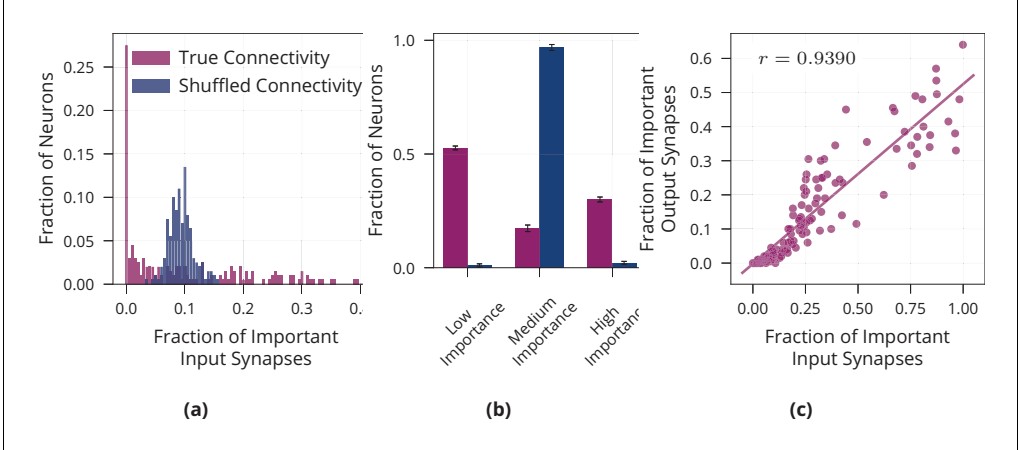

**Figure 3.** Neuron-level sparsity of network after learning. (**a**) Histogram of the fraction of important input synapses per neuron for second layer neurons after learning for true and randomly shuffled connectivity (see *Appendix 1—figure 2a* for other layers). (**b**) Same data as (**a**), showing number of low/medium/high importance neurons, where high/low importance neurons have at least two standard deviations more/less important inputs than the mean of random connectivity. (**c**) Scatter plot of first layer neurons showing the number of important input and output synapses after learning on MNIST, Pearson correlation is $r = 0.9390$ (see *Appendix 1—figure 2b* for other layers). Data in (**a**) and (**c**) are from one representative run, error bars in (**b**) show standard error over three repetitions.

count the number of highly important neurons, where we define a neuron to be highly important if its number of active inputs is two standard deviations below or above the mean (mean and standard deviation from shuffled baseline). We find that our model network with presynaptic stochasticity has disproportionate numbers of highly important and unimportant neurons, see *Figure 3b*. Moreover, we check whether neurons with many important inputs tend to have many important outputs, indeed finding a correlation of $r = 0.93$, see *Figure 3c*. These analyses all support the claim that the network is sparse not only on a synaptic but also on a neuronal level.

Finally, we investigate how release probabilities evolve from a theoretical viewpoint under the proposed learning rule. Note that the evolution of release probabilities is a random process, since it depends on the random input to the network. Under mild assumptions, we show (Materials and methods) that release probabilities are more likely to increase for synapses with large Fisher information (In this context, the Fisher information is a measure of sensitivity of the network to changes in synapses, measuring how important preserving a given synapse is for network function.). Thus, synapses with large release probabilities will tend to have high Fisher information. We validate this theoretical prediction empirically, see *Figure 2d*.

## Presynaptically driven consolidation helps alleviate the stability-plasticity dilemma

While the biological mechanisms addressing the stability-plasticity dilemma are diverse and not fully understood, it has been demonstrated experimentally that preserving memories requires maintaining the synapses which encode these memories (*Yang et al., 2009*; *Hayashi-Takagi et al., 2015*; *Cichon and Gan, 2015*). In this context, theoretical work suggests that the Fisher information is a useful way to quantify which synapses should be maintained (*Kirkpatrick et al., 2017*). Inspired by these insights, we hypothesise that the synaptic importance encoded in release probabilities can be used to improve the network's memory retention by selectively stabilising important synapses.

We formalise this hypothesis in our model by lowering the learning rate (plasticity) of synapses according to their importance (release probability). Concretely, the learning rate $\eta = \eta(p_i)$ used in (3) is scaled as follows

$$\eta(p_i) = \eta_0 \cdot (1 - p_i). \tag{5}$$

such that the learning rate is smallest for important synapses with high release probability. $\eta_0$

denotes a base learning rate that is shared by all synapses. We complement this consolidation mechanism by freezing the presynaptic release probabilities $p_i$ once they have surpassed a predefined threshold $p_{\text{freeze}}$. This ensures that a synapse whose presynaptic release probability was high for a previous task retains its release probability even when unused during consecutive tasks. In other words, the effects of presynaptic long-term depression (LTD) are assumed to act on a slower timescale than learning single tasks. Note that the freezing mechanism ensures that all synaptic strengths $\bar{w}_i$ retain a small degree of plasticity, since the learning rate modulation factor $(1 - p_i)$ remains greater than 0.

To test our hypothesis that presynaptically driven consolidation allows the network to make improved stability-plasticity trade-offs, we sequentially present a number of tasks and investigate the networks behaviour. We mainly focus our analysis on a variation of the MNIST handwritten digit dataset, in which the network has to successively learn the parity of pairs of digits, see *Figure 4a*. Additional experiments are reported in the Appendix, see *Appendix 1—table 1*.

First, we investigate whether presynaptic consolidation improves the model's ability to remember old tasks. To this end, we track the accuracy on the first task over the course of learning, see *Figure 4b*. As a control, we include a model without consolidation and with deterministic synapses. While both models learn the first task, the model without consolidation forgets more quickly, suggesting that the presynaptic consolidation mechanism does indeed improve memory.

Next, we ask how increased stability interacts with the network's ability to remain plastic and learn new tasks. To assess the overall trade-off between stability and plasticity, we report the average accuracy over all five tasks, see *Figure 4c*.

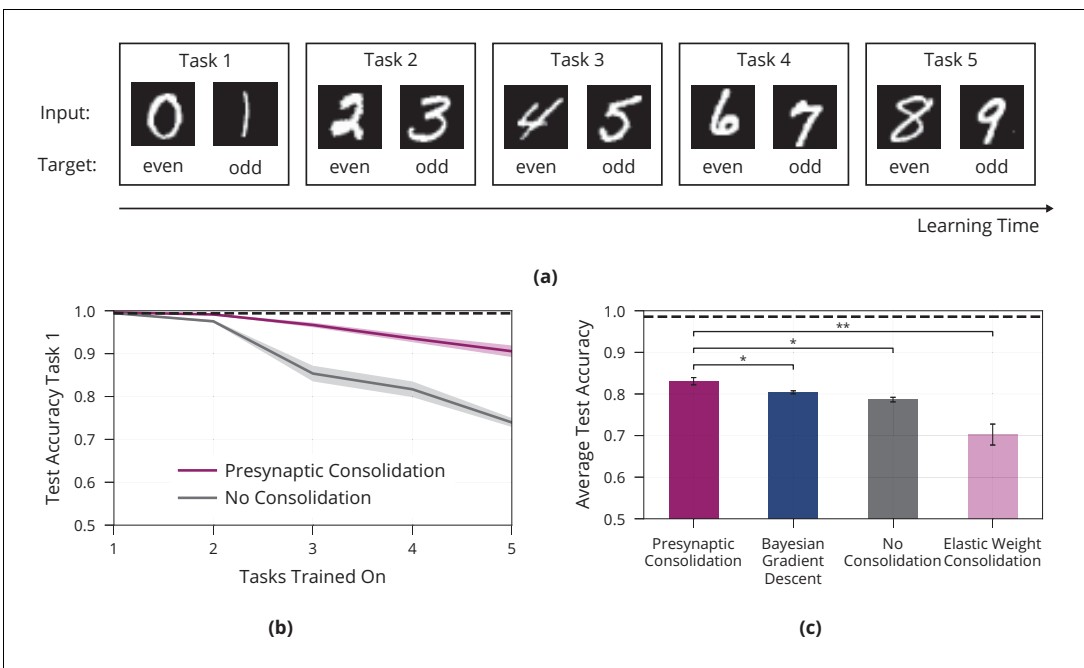

**Figure 4.** Lifelong learning in a model with presynaptically driven consolidation. (a) Schematic of the lifelong learning task Split MNIST. In the first task the model network is presented 0 s and 1 s, in the second task it is presented 2 s and 3 s, etc. For each task the model has to classify the inputs as even or odd. At the end of learning, it should be able to correctly classify the parity of all digits, even if a digit has been learned in an early task. (b) Accuracy of the first task when learning new tasks. Consolidation leads to improved memory preservation. (c) Average accuracies of all learned tasks. The presynaptic consolidation model is compared to a model without consolidation and two state-of-the-art machine learning algorithms. Differences to these models are significant in independent t-tests with either $p<0.05$ (marked with *) or with $p<0.01$ (marked with **). Dashed line indicates an upper bound for the network's performance, obtained by training on all tasks simultaneously. Panels (b) and (c) show accumulated data for three repetitions of the experiment. Shaded regions in (b) and error bars in (c) show standard error.

We find that the presynaptic consolidation model performs better than a standard model with deterministic synapses and without consolidation. In addition, we compare performance to two state-of-the art machine learning algorithms: The well-known algorithm Elastic Weight Consolidation (EWC) (*Kirkpatrick et al., 2017*) explicitly relies on the Fisher information and performs a separate consolidation phase after each task. Bayesian Gradient Descent (BGD) (*Zeno et al., 2018*) is a Bayesian approach that models synapses as distributions, but does not capture the discrete nature of synaptic transmission. The presynaptic consolidation mechanism performs better than both these state-of-the-art machine learning algorithms, see *Figure 4c*. Additional experiments in the Appendix suggest overall similar performance of Presynaptic Consolidation to BGD and similar or better performance than EWC.

To determine which components of our model contribute to its lifelong learning capabilities, we perform an ablation study, see *Figure 5a*. We aim to separate the effect of (1) consolidation mechanisms and (2) presynaptic plasticity.

First, we remove the two consolidation mechanisms, learning rate modulation and freezing release probabilities, from the model with stochastic synapses. This yields a noticeable decrease in performance during lifelong learning, thus supporting the view that stabilising important synapses contributes to addressing the stability-plasticity dilemma.

Second, we aim to disentangle the effect of presynaptic plasticity from the consolidation mechanisms. We therefore introduce a control in which presynaptic plasticity but not consolidation is blocked. Concretely, the control has 'ghost release probabilities' $\tilde{p}_i$ evolving according to *Equation (4)* and modulating plasticity according to *Equation (5)*; but the synaptic release probability is fixed at 0.5. We see that this control performs worse than the original model with a drop in accuracy of 1.4 on Split MNIST ($t = 3.44$, $p<0.05$) and a drop of accuracy of 5.6 on Permuted MNIST ($t = 6.72, p<0.01$). This suggests that presynaptic plasticity, on top of consolidation, helps to stabilise the network. We believe that this can be attributed to the sparsity induced by the presynaptic plasticity which decreases overlap between different tasks.

The above experiments rely on a gradient-based learning rule for multilayer perceptrons. To test whether presynaptic consolidation can also alleviate stability-plasticity trade-offs in other settings, we study its effects on learning in a standard perceptron (*Rosenblatt, 1958*). We train the perceptron sequentially on five pattern memorisation tasks, see Materials and methods for full details. We find that the presynaptically consolidated perceptron maintains a more stable memory of the first

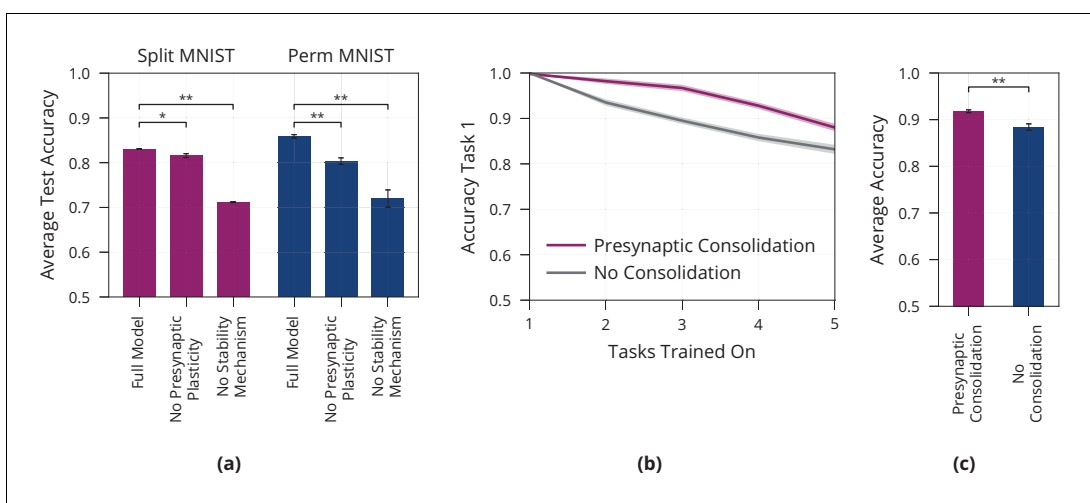

**Figure 5.** Model ablation and lifelong learning in a standard perceptron. (a) Ablation of the Presynaptic Consolidation model on two different lifelong learning tasks, see full text for detailed description. Both presynaptic plasticity and synaptic stabilisation significantly improve memory. (b+c) Lifelong Learning in a Standard Perceptron akin to *Figure 4b,c*, showing the accuracy of the first task when learning consecutive tasks in (b) as well as the average over all five tasks after learning all tasks in (c). Error bars and shaded regions show standard error of three respectively ten repetitions, in (a), respectively (b+c). All pair-wise comparisons are significant, independent t-tests with $p<0.01$ (denoted by **) or with $p<0.05$ (denoted by *).

task, see *Figure 5b*. In addition, this leads to an overall improved stability-plasticity trade-off, see *Figure 5c* and shows that the effects of presynaptic consolidation in our model extend beyond gradient-based learning.

## Discussion

### Main contribution

Information transmission in synapses is stochastic. While previous work has suggested that stochasticity allows to maximise the amount of information transmitted per unit of energy spent, this analysis has been restricted to single synapses. We argue that the relevant quantity to be considered is task-dependent information transmitted by entire networks. Introducing a simple model of the all-or-nothing nature of synaptic transmission, we show that presynaptic stochasticity enables networks to allocate energy more efficiently. We find theoretically as well as empirically that learned release probabilities encode the importance of weights for network function according to the Fisher information. Based on this finding, we suggest a novel computational role for presynaptic stochasticity in lifelong learning. Our experiments provide evidence that coupling information encoded in the release probabilities with modulated plasticity can help alleviate the stability-plasticity dilemma.

### Modelling assumptions and biological plausibility

Stochastic synaptic transmission

Our model captures the occurrence of synaptic failures by introducing a Bernoulli random variable governing whether or not neurotransmitter is released. Compared to classical models assuming deterministic transmission, this is one step closer to experimentally observed binomial transmission patterns, which are caused by multiple, rather than one, release sites between a given neuron and dendritic branch. Importantly, our simplified model accounts for the event that there is no postsynaptic depolarisation at all. Even in the presence of multiple release sites, this event has non-negligible probability: Data from cultured hippocampal neurons (*Branco et al., 2008*, *Figure 2D*) and the neocortex (*Hardingham et al., 2010*, *Appendix 1—figure 2c*) shows that the probability $(1-p)^N$ that none of $N$ release sites with release probability $p$ is active, is around 0.3–0.4 even for $N$ as large as 10. More recent evidence suggests an even wider range of values depending on the extracellular calcium concentration (*Sakamoto et al., 2018*).

Presynaptic long-term plasticity

A central property of our model builds on the observation that the locus of expression for long-term plasticity can both be presynaptic and postsynaptic (*Larkman et al., 1992*; *Lisman and Raghavachari, 2006*; *Bayazitov et al., 2007*; *Sjöström et al., 2007*; *Bliss and Collingridge, 2013*; *Costa et al., 2017*). The mechanisms to change either are distinct and synapse-specific (*Yang and Calakos, 2013*; *Castillo, 2012*), but how exactly pre- and postsynaptic forms of long-term potentiation (LTP) and long-term depression (LTD) interact is not yet fully understood (*Monday et al., 2018*). The induction of long-term plasticity is thought to be triggered postsynaptically for both presynaptic and postsynaptic changes (*Yang and Calakos, 2013*; *Padamsey and Emptage, 2014*) and several forms of presynaptic plasticity are known to require retrograde signalling (*Monday et al., 2018*), for example through nitric oxide or endocannabinoids (*Heifets and Castillo, 2009*; *Andrade-Talavera et al., 2016*; *Costa et al., 2017*). This interaction between pre- and postsynaptic sites is reflected by our learning rule, in which both pre- and postsynaptic changes are governed by postsynaptic updates and require communication between pre- and postsynapse. The proposed presynaptic updates rely on both presynaptic LTP and presynaptic LTD. At least one form of presynaptic long-term plasticity is known to be bidirectional switching from potentiation to depression depending on endocannabinoid transients (*Cui et al., 2015*; *Cui et al., 2016*).

Link between presynaptic release and synaptic stability

Our model suggests that increasing the stability of synapses with large release probability improves memory. Qualitatively, this is in line with observations that presynaptic boutons, which contain

stationary mitochondria (*Chang et al., 2006*; *Obashi and Okabe, 2013*), are more stable than those which do not, both on short (*Sun et al., 2013*) and long timescales of at least weeks (*Lees et al., 2019*). Quantitatively, we find evidence for such a link by re-analysing data (Data was made publicly available in *Costa et al., 2017*). from *Sjöström et al., 2001* for a spike-timing-dependent plasticity protocol in the rat primary visual cortex: *Appendix 1—figure 4* shows that synapses with higher initial release probability are more stable than those with low release probabilities for both LTP and LTD.

## Credit assignment

In our multilayer perceptron model, updates are computed using backpropagated gradients. Whether credit assignment in the brain relies on backpropagation – or more generally gradients – remains an active area of research, but several alternatives aiming to increase biological plausibility exist and are compatible with our model (*Sacramento et al., 2018*; *Lillicrap et al., 2016*; *Lee et al., 2015*). To check that the proposed mechanism can also operate without gradient information, we include an experiment with a standard perceptron and its gradient-free learning rule (*Rosenblatt, 1958*), see *Figure 5b, c*.

## Correspondence to biological networks

We study general rate-based neural networks raising the question in which biological networks or contexts one might expect the proposed mechanisms to be at work. Our experiments suggest that improved energy efficiency can at least partly be attributed to the sparsification induced by presynaptic stochasticity (cf. *Olshausen and Field, 2004*). Networks which are known to rely on sparse representations are thus natural candidates for the dynamics investigated here. This includes a wide range of sensory networks (*Perez-Orive et al., 2002*; *Hahnloser et al., 2002*; *Crochet et al., 2011*; *Quiroga et al., 2005*) as well as areas in the hippocampus (*Wixted et al., 2014*; *Lodge and Bischofberger, 2019*).

In the context of lifelong learning, our learning rule provides a potential mechanism that helps to slowly incorporate new knowledge into a network with preexisting memories. Generally, the introduced consolidation mechanism could benefit the slow part of a complementary learning system as proposed by *McClelland et al., 1995*; *Kumaran et al., 2016*. Sensory networks in particular might utilize such a mechanism as they require to learn new stimuli while retaining the ability to recognise previous ones (*Buonomano and Merzenich, 1998*; *Gilbert et al., 2009*; *Moczulska et al., 2013*). Indeed, in line with the hypothesis that synapses with larger release probability are more stable, it has been observed that larger spines in the mouse barrel cortex are more stable. Moreover, novel experiences lead to the formation of new, stable spines, similar to our findings reported in *Appendix 1—figure 3b*.

## Related synapse models

### Probabilistic synapse models

The goal of incorporating and interpreting noise in models of neural computation is shared by many computational studies. Inspired by a Bayesian perspective, neural variability is often interpreted as representing uncertainty (*Ma et al., 2006*; *Fiser et al., 2010*; *Kappel et al., 2015*; *Haefner et al., 2016*), or as a means to prevent overfitting (*Wan et al., 2013*). The Bayesian paradigm has been applied directly to variability of individual synapses in neuroscience (*Aitchison et al., 2014*; *Aitchison and Latham, 2015*; *Aitchison et al., 2021*) and machine learning (*Zeno et al., 2018*). It prescribes decreasing the plasticity of synapses with low posterior variance. A similiar relationship can be shown to hold for our model as described in the Material and Methods. In contrast to common Bayesian interpretations (*Zeno et al., 2018*; *Aitchison and Latham, 2015*; *Kappel et al., 2015*) which model release statistics as Gaussians and optimise complex objectives (see also *Llera-Montero et al., 2019*) our simple proposal represents the inherently discrete nature of synaptic transmission more faithfully.

### Complex synapse models

In the context of lifelong learning, our model's consolidation mechanism is similar to Elastic Weight Consolidation (EWC) (*Kirkpatrick et al., 2017*), which explicitly relies on the Fisher information to

consolidate synapses. Unlike EWC, our learning rule does not require a task switch signal and does not need a separate consolidation phase. Moreover, our model can be interpreted as using distinct states of plasticity to protect memories. This general idea is formalised and analysed thoroughly by theoretical work on cascade models of plasticity (*Fusi et al., 2005*; *Roxin and Fusi, 2013*; *Benna and Fusi, 2016*). The resulting model (*Benna and Fusi, 2016*) has also been shown to be effective in lifelong learning settings (*Kaplanis et al., 2018*).

### Synaptic importance may govern energy-information trade-offs

Energy constraints are widely believed to be a main driver of evolution (*Niven and Laughlin, 2008*). From brain size (*Isler and van Schaik, 2009*; *Navarrete et al., 2011*), to wiring cost (*Chen et al., 2006*), down to ion channel properties (*Alle et al., 2009*; *Sengupta et al., 2010*), presynaptic transmitter release (*Savtchenko et al., 2013*) and postsynaptic conductance (*Harris et al., 2015*), various components of the nervous system have been shown to be optimal in terms of their total metabolic cost or their metabolic cost per bit of information transmitted.

Crucially, there is evidence that the central nervous system operates in varying regimes, making different trade-offs between synaptic energy demand and information transmission: *Perge et al., 2009*; *Carter and Bean, 2009*; *Hu and Jonas, 2014* all find properties of the axon (thickness, sodium channel properties), which are suboptimal in terms of energy per bit of information. They suggest that these inefficiencies occur to ensure fast transmission of highly relevant information.

We propose that a similar energy/information trade-off could govern network dynamics preferentially allocating more energy to the most relevant synapses for a given task. Our model relies on a simple, theoretically justified learning rule to achieve this goal and leads to overall energy savings. Neither the trade-off nor the overall savings can be accounted for by previous frameworks for energy-efficient information transmission at synapses (*Levy and Baxter, 2002*; *Harris et al., 2012*).

This view of release probabilities and related metabolic cost provides a way to make the informal notion of 'synaptic importance' concrete by measuring how much energy is spent on a synapse. Interestingly, our model suggests that this notion is helpful beyond purely energetic considerations and can in fact help to maintain memories during lifelong learning.

## Materials and methods

### Summary of learning rule

Our learning rule has two components, an update for the presynaptic release probability $p_i$ and an update for the postsynaptic strength $m_i$. The update of the synaptic strength $m_i$ is defined implicitly through updating the expected synaptic strength $\bar{w}$

$$\bar{w}_i^{(t+1)} = \bar{w}_i^{(t)} - \eta g_i, \quad \text{where} \quad g_i = \frac{\partial L(\bar{w}^{(t)}, p^{(t)})}{\partial \bar{w}_i^{(t)}} \tag{6}$$

and the presynaptic update is given by

$$p_i^{(t+1)} = \begin{cases} p_i^{(t)} + p_{\text{up}}, & \text{if} |g_i| > g_{\text{lim}}, \\ p_i^{(t)} - p_{\text{down}}, & \text{if} |g_i| \leq g_{\text{lim}}. \end{cases} \tag{7}$$

This leads to the following explicit update rule for the synaptic strength $m_i = \frac{\bar{w}_i}{p_i}$

$$m_i^{(t+1)} = \frac{1}{p_i^{(t+1)}} \left( p_i^{(t)} m_i^{(t)} - \eta g_i \right) \tag{8}$$

$$= \frac{p_i^{(t)}}{p_i^{(t+1)}} m_i^{(t)} - \frac{\eta}{p_i^{(t+1)} p_i^{(t)}} \frac{\partial L(m^{(t)}, p^{(t)})}{\partial m_i^{(t)}} \tag{9}$$

where we used the chain rule to rewrite $g_i = \frac{\partial L}{\partial \bar{w}_i} = \frac{\partial L}{\partial m_i} \cdot \frac{\partial m_i}{\partial \bar{w}_i} = \frac{\partial L}{\partial m_i} \cdot \frac{1}{p_i}$. For the lifelong learning experiment, we additionally stabilise high release probability synapses by multiplying the learning rate by $(1 - p_i)$

for each synapse and by freezing release probabilities (but not strengths) when they surpass a predefined threshold $p_{\text{freeze}}$.

## Theoretical analysis of presynaptic learning rule

As indicated in the results section the release probability $p_i$ is more likely to be large when the Fisher information of the synaptic strength $w_i$ is large as well. This provides a theoretical explanation to the intuitive correspondence between release probability and synaptic importance. Here, we formalise this link starting with a brief review of the Fisher information.

### Fisher information

The Fisher information is a measure for the networks sensitivity to changes in parameters. Under additional assumptions it is equal to the Hessian of the loss function (**Pascanu and Bengio, 2013**; **Martens, 2014**), giving an intuitive reason why synapses with high Fisher information should not be changed much if network function is to be preserved.

Formally, for a model with parameter vector $\theta$ predicting a probability distribution $f_\theta(X, y)$ for inputs $X$ and labels $y$ drawn from a joint distribution $\mathcal{D}$, the Fisher information matrix is defined as

$$\mathbb{E}_{X \sim \mathcal{D}} \mathbb{E}_{y \sim f_\theta(y \mid X)} \Big[ \Big( \frac{\partial \ln f_\theta(X, y)}{\partial \theta} \Big) \Big( \frac{\partial \ln f_\theta(X, y)}{\partial \theta} \Big)^T \Big].$$

Note that this expression is independent of the actual labels $y$ of the dataset and that instead we sample labels from the model's predictions. If the model makes correct predictions, we can replace the second expectation, which is over $y \sim f_\theta(y \mid X)$, by the empirical labels $y$ of the dataset for an approximation called the Empirical Fisher information. If we further only consider the diagonal entries – corresponding to a mean-field approximation – and write $g_i(X, y) = \frac{\partial \ln f_\theta(X, y)}{\partial \theta_i}$ we obtain the following expression for the $i$-th entry of the diagonal Empirical Fisher information:

$$F_i = \mathbb{E}_{X, y \sim \mathcal{D}}[g_i(X, y)^2].$$

Note that this version of the Fisher information relies on the same gradients that are used to update the parameters of the multilayer perceptron, see **Equations (3), (4)**.

Under the assumption that the learned probability distribution $f(\cdot \mid X, \theta)$ equals the real probability distribution, the Fisher information equals the Hessian of the cross entropy loss (i.e. the negative log-probabilities) with respect to the model parameters (**Pascanu and Bengio, 2013**; **Martens, 2014**). The Fisher information was previously used in machine learning to enable lifelong learning (**Kirkpatrick et al., 2017**; **Huszár, 2018**) and it has been shown that other popular lifelong learning methods implicitly rely on the Fisher information (**Benzing, 2020**).

### Link between release probabilities and Fisher information

We now explain how our learning rule for the release probability is related to the Fisher information. For simplicity of exposition, we focus our analysis on a particular sampled subnetwork with deterministic synaptic strengths. Recall that update rule (4) for release probabilities increases the release probability, if the gradient magnitude $|g_i|$ is above a certain threshold, $g_i > |g_{\text{lim}}|$, and decreases them otherwise. Let us denote by $p_i^+$ the probability that the $i$-th release probability is increased. Then

$$p_i^+ := \Pr[|g_i| > g_{\text{lim}}] = \Pr[g_i^2 > g_{\text{lim}}^2], \tag{10}$$

where the probability space corresponds to sampling training examples. Note that $\mathbb{E}[g_i^2] = F_i$ by definition of the Empirical Fisher information $F_i$. So if we assume that $\Pr[g_i^2 > g_{\text{lim}}^2]$ depends monotonically on $\mathbb{E}[g_i^2]$ then we already see that $p_i^+$ depends monotonically on $F_i$. This in turn implies that synapses with a larger Fisher information are more likely to have a large release probability, which is what we claimed. We now discuss the assumption made above.

## Assumption: $\Pr[g_i^2 > g_{\lim}^2]$ depends monotonically on $\mathbb{E}[g_i^2]$

While this assumption is not true for arbitrary distributions of $g$, it holds for many commonly studied parametric families and seems likely to hold (approximately) for realistic, non-adversarially chosen distributions. For example, if each $g_i$ follows a normal distribution $g_i \sim \mathcal{N}(\mu_i, \sigma_i^2)$ with varying $\sigma_i$ and $\sigma_i \gg \mu_i$, then

$$F_i = \mathbb{E}[g_i^2] \approx \sigma_i^2$$

and

$$p_i^+ = \Pr[g_i^2 > g_{\lim}^2] \approx \operatorname{erfc}\left(\frac{g_{\lim}}{\sigma_i \sqrt{2}}\right)$$

so that $p_i^+$ is indeed monotonically increasing in $F_i$. Similar arguments can be made for example for a Laplace distribution, with scale larger than mean.

## Link between learning rate modulation and Bayesian updating

Recall that we multiply the learning rate of each synapse by $(1 - p_i)$, see *Equation (5)*. This learning rate modulation can be related to the update prescribed by Bayesian modelling. As shown before, synapses with large Fisher information tend to have large release probability, which results in a decrease of the plasticity of synapses with large Fisher information. We can treat the (diagonal) Fisher information as an approximation of the posterior precision based on a Laplace approximation of the posterior likelihood (*Kirkpatrick et al., 2017*) which exploits that the Fisher information approaches the Hessian of the loss as the task gets learned (*Martens, 2014*). Using this relationship, our learning rate modulation tends to lower the learning rate of synapses with low posterior variance as prescribed by Bayesian modelling.

## Practical approximation

The derivation above assumes that each gradient $g$ is computed using a single input, so that $\mathbb{E}[g^2]$ equals the Fisher information. While this may be the biologically more plausible setting, in standard artificial neural network (ANN) training the gradient is averaged across several inputs (mini-batches). Despite this modification, $g^2$ remains a good, and commonly used, approximation of the Fisher, see for example *Khan et al., 2018*; *Benzing, 2020*.

## Perceptron for lifelong learning

To demonstrate that our findings on presynaptic stochasticity and plasticity are applicable to other models and learning rules, we include experiments for the standard perceptron (*Rosenblatt, 1958*) in a lifelong learning setting.

### Model

The perceptron is a classical model for a neuron with multiple inputs and threshold activation function. It is used to memorise the binary labels of a number of input patterns where input patterns are sampled uniformly from $\{-1, 1\}^N$ and their labels are sampled uniformly from $\{-1, 1\}$. Like in ANNs, the output neuron of a perceptron computes a weighted sum of its inputs followed by nonlinear activation $\sigma(\cdot)$:

$$\underbrace{a^{\text{post}}}_{\substack{\text{postsynaptic} \\ \text{activation}}} = \sigma\Big(\sum_{i=1}^{n} w_i \underbrace{a_i^{\text{pre}}}_{\substack{\text{i-th presynaptic} \\ \text{input}}}\Big). \tag{11}$$

The only difference to the ANN model is that the nonlinearity is the sign function and that there is only one layer. We model each synapse $w_i$ as a Bernoulli variable $r_i$ with synaptic strength $m_i$ and release probability $p_i$ just as before, see *Equation (1)*. The expected strengths $\bar{w}_i$ are learned according to the standard perceptron learning rule (*Rosenblatt, 1958*). The only modification we make is averaging weight updates across 5 inputs, rather than applying an update after each input. Without this modification, the update size $g_i$ for each weight $w_i$ would be constant according to the

perceptron learning rule. Consequently, our update rule for $p_i$ would not be applicable. However, after averaging across five patterns we can apply the same update rule for $p_i$ as previously, see *Equation (4)*, and also use the same learning rate modification, see *Equation (5)*. We clarify that $g_i$ now refers to the update of expected strength $\bar{w}_i$. In the case of ANN this is proportional to the gradient, while in the case of the non-differentiable perceptron it has no additional interpretation.

## Experiments

For the lifelong learning experiments, we used five tasks, each consisting of 100 randomly sampled and labelled patterns of size $N = 1000$. We compared the perceptron with learned stochastic weights to a standard perceptron. For the standard perceptron, we also averaged updates across five patterns. Both models were sequentially trained on five tasks, using 25 passes through the data for each task.

We note that for more patterns, when the perceptron gets closer to its maximum capacity of $2N$, the average accuracies of the stochastic and standard perceptron become more similar, suggesting that the benefits of stochastic synapses occur when model capacity is not fully used.

As metaplasticity parameters we used $g_{\text{lim}} = 0.1, p_{\text{up}} = p_{\text{down}} = 0.2$ and $p_{\text{min}} = 0.25, p_{\text{freeze}} = 0.9$. These were coarsely tuned on an analogous experiment with only two tasks instead of five.

## Experimental setup

### Code availability

Code for all experiments is publicly available at github.com/smonsays/presynaptic-stochasticity (*Schug, 2021*, copy archived at swh:1:rev:de0851773cd1375b885dcdb18e711a2fb6eb06a4).

### Metaplasticity parameters

Our method has a number of metaplasticity parameters, namely $p_{\text{up}}$, $p_{\text{down}}$, $g_{\text{lim}}$ and the learning rate $\eta$. For the lifelong learning experiments, there is an additional parameter $p_{\text{freeze}}$. For the energy experiments, we fix $p_{\text{up}} = p_{\text{down}} = 0.07$, $g_{\text{lim}} = 0.001$ and choose $\eta = 0.05$ based on coarse, manual tuning. For the lifelong learning experiments, we choose $\eta_0 \in \{0.01, 0.001\}$ and optimise the remaining metaplasticity parameters through a random search on one task, namely Permuted MNIST, resulting in $p_{\text{up}} = 0.0516$, $p_{\text{down}} = 0.0520$ and $g_{\text{lim}} = 0.001$. We use the same fixed parametrisation for all other tasks, namely Permuted Fashion MNIST, Split MNIST and Split Fashion MNIST (see below for detailed task descriptions). For the ablation experiment in *Figure 5a*, metaplasticity parameters were re-optimised for each ablation in a random search to ensure a fair, meaningful comparison.

### Model robustness

We confirmed that the model is robust with respect to the exact choice of parameters. For the energy experiments, de- or increasing $p_{\text{up}}, p_{\text{down}}$ by 25 does not qualitatively change results.

For the lifelong learning experiment, the chosen tuning method is a strong indicator of robustness: The metaplasticitiy parameters are tuned on one setup (Permuted MNIST) and then transferred to others (Split MNIST, Permuted and Split Fashion MNIST). The results presented in *Appendix 1— table 1* show that the parameters found in one scenario are robust and carry over to several other settings. We emphasise that the differences between these scenarios are considerable. For example, for permuted MNIST consecutive input distributions are essentially uncorrelated by design, while for Split (Fashion) MNIST input distributions are strongly correlated. In addition, from MNIST to Fashion MNIST the number of 'informative' pixels changes drastically.

### Lifelong learning tasks

For the lifelong learning experiments, we tested our method as well as baselines in several scenarios on top of the Split MNIST protocol described in the main text.

## Permuted MNIST

In the Permuted MNIST benchmark, each task consists of a random but fixed permutation of the input pixels of all MNIST images (*Goodfellow et al., 2013*). We generate 10 tasks using this procedure and present them sequentially without any indication of task boundaries during training. A main reason to consider the Permuted MNIST protocol is that it generates tasks of equal difficulty.

### Permuted and split fashion MNIST

Both the Split and Permuted protocol can be applied to other datasets. We use them on the Fashion MNIST dataset (*Xiao et al., 2017*) consisting of 60,000 greyscale images of 10 different fashion items with $28 \times 28$ pixels.

### Continuous permuted MNIST

We carry out an additional experiment on the continuous Permuted MNIST dataset (*Zeno et al., 2018*). This is a modified version of the Permuted MNIST dataset which introduces a smooth transition period between individual tasks where data from both distributions is mixed. It removes the abrupt change between tasks and allows us to investigate if our method depends on such an implicit task switch signal. We observe a mean accuracy over all tasks of $0.8539 \pm 0.006$ comparable to the non-continuous case suggesting that our method does not require abrupt changes from one task to another.

## Neural network training

Our neural network architecture consists of two fully connected hidden layers of 200 neurons without biases with rectified linear unit activation functions $\sigma(x)$. The final layer uses a softmax and cross-entropy loss. Network weights were initialised according to the PyTorch default for fully connected layers, which is similar to Kaiming uniform initialisation (*Glorot and Bengio, 2010*; *He et al., 2015*) but divides weights by an additional factor of $\sqrt{6}$. We use standard stochastic gradient descent to update the average weight $\bar{w}_i$ only altered by the learning rate modulation described for the lifelong learning experiments. We use a batch size of 100 and train each task for 10 epochs in the lifelong learning setting. In the energy-information experiments we train the model for 50 epochs.

## Acknowledgements

We thank João Sacramento and Mark van Rossum for stimulating discussions and helpful comments.

## Additional information

### Funding

| Funder | Grant reference number | Author |
|---|---|---|
| Swiss National Science Foundation | PZ00P318602 | Simon Schug |
| Swiss National Science Foundation | CRSII5_173721 | Frederik Benzing Angelika Steger |

The funders had no role in study design, data collection and interpretation, or the decision to submit the work for publication.

### Author contributions

Simon Schug, Frederik Benzing, Conceptualization, Data curation, Software, Formal analysis, Validation, Investigation, Visualization, Methodology, Writing - original draft, Writing - review and editing; Angelika Steger, Conceptualization, Resources, Data curation, Supervision, Funding acquisition, Visualization, Writing - review and editing

### Author ORCIDs

Simon Schug (iD) https://orcid.org/0000-0002-5305-2547
Frederik Benzing (iD) https://orcid.org/0000-0001-6580-8690

### Decision letter and Author response

Decision letter https://doi.org/10.7554/eLife.69884.sa1
Author response https://doi.org/10.7554/eLife.69884.sa2

## Additional files

### Supplementary files
- Source code 1. Code for presynaptic stochasticity model and baselines.
- Source data 1. Raw data and code to create figures.
- Transparent reporting form

### Data availability

Code for experiments is part of the submission and is published on GitHub (https://github.com/smonsays/presynaptic-stochasticity copy archived at https://archive.softwareheritage.org/swh:1:rev:de0851773cd1375b885dcdb18e711a2fb6eb06a4).

The following previously published dataset was used:

| Author(s) | Year | Dataset title | Dataset URL | Database and Identifier |
|---|---|---|---|---|
| Costa R, Froemke R, Sjöström P, van Rossum M | 2015 | Data from: Unified pre- and postsynaptic long-term plasticity enables reliable and flexible learning | https://doi.org/10.5061/dryad.p286g | Dryad Digital Repository, 10.5061/dryad.p286g |

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

## Appendix 1

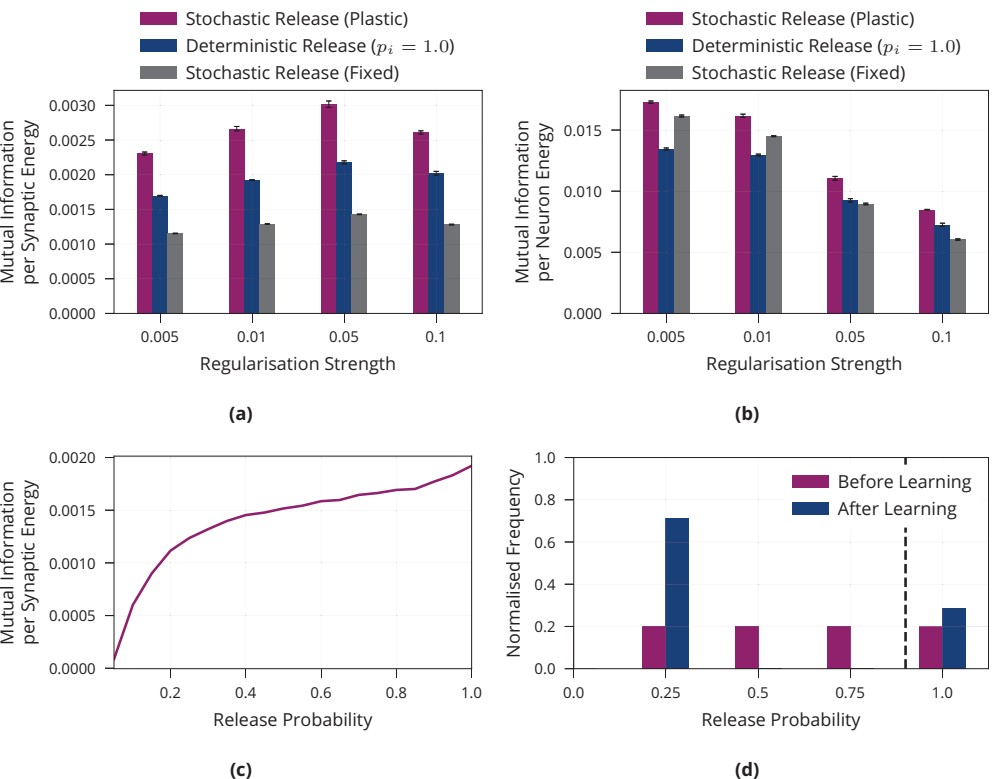

**Appendix 1—figure 1.** Additional results on energy efficiency of model with stochastic and plastic release. (**a**) Mutual information per energy analogous to *Figure 1b*, but showing results for different regularisation strengths rather than the best result for each model. As described in the main part, energy is measured via its synaptic contribution. (**b**) Same experiment as in (**a**) but energy is measured as the metabolic cost incurred by the activity of neurons by calculating their average rate of activity. (**c**) Maximum mutual information per energy for a multilayer perceptron with fixed release probability and constant regularisation strength of 0.01. This is the same model as 'Stochastic Release (Fixed)' in (**a**), but for a range of different values for the release probability. This is in line with the single synapse analysis in *Harris et al., 2012*. For each model, we searched over different learning rates and report the best result. (**d**) Analogous to *Figure 2a*, but release probabilities were initialised independently, uniformly at random in the interval $[0.25, 1]$ rather than with a fixed value of 0.25. Error bars in (**a**) and (**b**) denote the standard error for three repetitions of the experiment. (**c**) shows the best performing model for each release probability after a grid search over the learning rate. (**d**) shows aggregated data over three repetitions of the experiment.

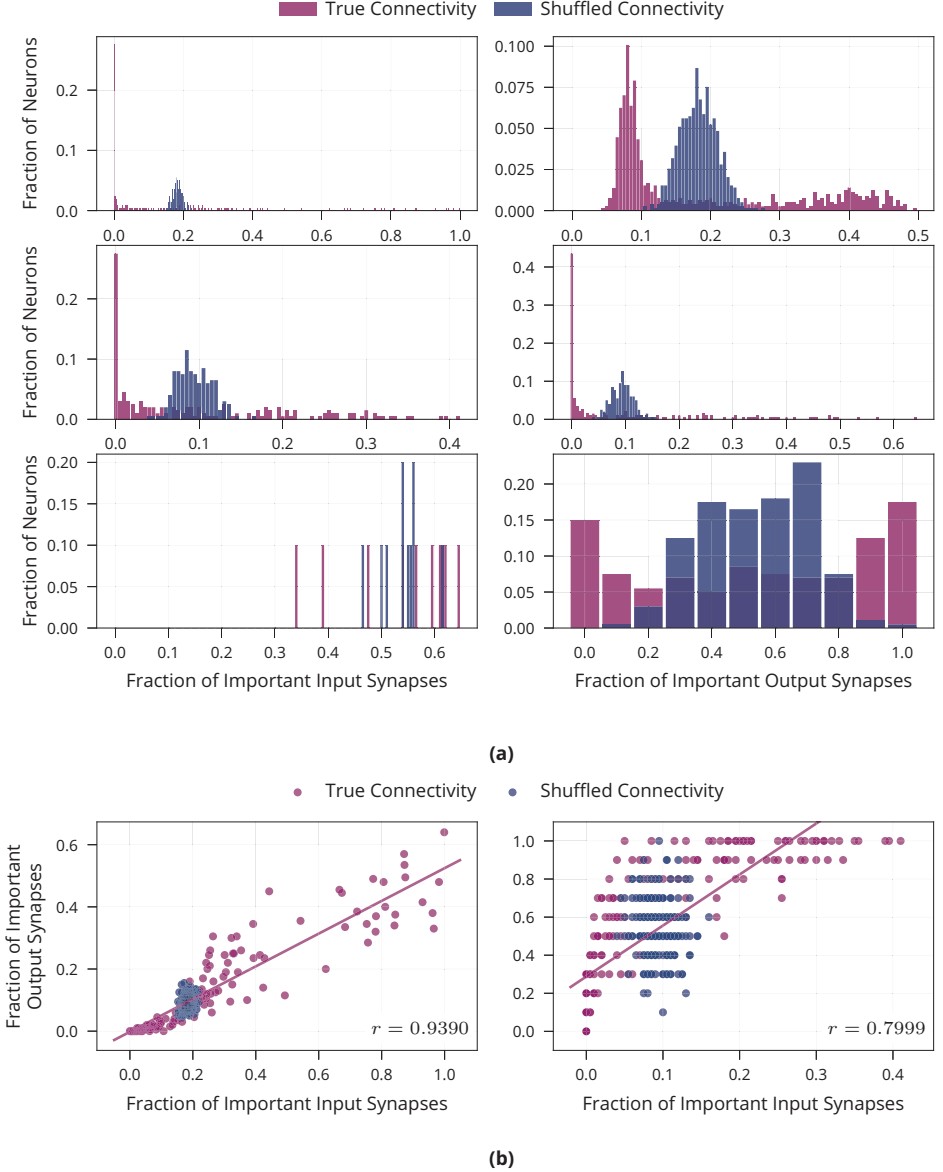

**Appendix 1—figure 2.** Additional results on neuron-level sparsity of network after learning. (**a**) Number of important synapses per neuron for all layers after learning on MNIST. The $i$-th row shows data from the $i$-th weight matrix of the network and we compare true connectivity to random connectivity. Two-sample Kolmogorov-Smirnov tests comparing the distribution of important synapses in the shuffled and unaltered condition are significant for all layers ($p<0.01$) except for the output neurons in the last layer (lower-left panel) ($p=0.41$). This is to be expected as all 10 output neurons in the last layer should be equally active and thus receive similar numbers of active inputs. (**b**) Scatter plot showing the number of important input and output synapses per neuron for both hidden layers after learning on MNIST. First hidden layer (left) has a Pearson correlation coefficient of $r=0.9390$. Second hidden layer (right) has a Pearson correlation coefficient of $r=0.7999$. Data is from one run of the experiment.

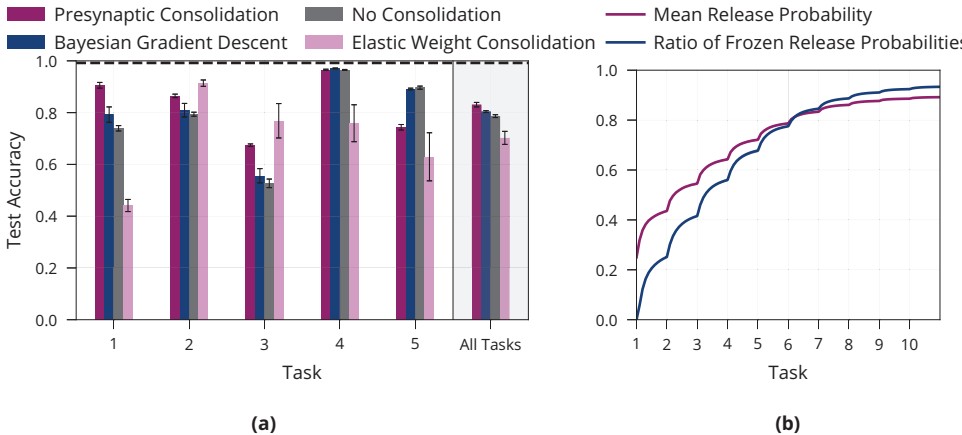

**Appendix 1—figure 3.** Additional results on lifelong learning in a model with presynaptically driven consolidation. (**a**) Detailed lifelong-learning results of various methods on Split MNIST, same underlying experiment as in *Figure 4c*. We report the test accuracy on each task of the final model (after learning all tasks). Error bars denote the standard error for three repetitions of the experiment. (**b**) Mean release probability and percentage of frozen weights over the course of learning ten permuted MNIST tasks. Error bars in (**a**) and shaded regions in (**b**) show standard error over three repetitions of the experiment.

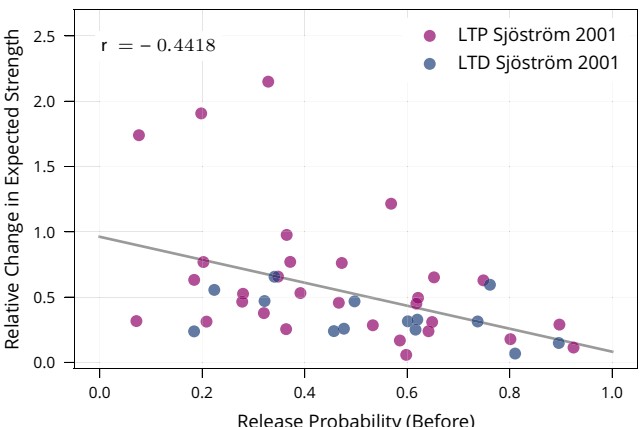

**Appendix 1—figure 4.** Biological evidence for stability of synapses with high release probability. To test whether synapses with high release probability are more stable than synapses with low release probability as prescribed by our model, we re-analysed data of *Sjöström et al., 2001* from a set of spike-timing-dependent plasticity protocols. The protocols induce both LTP and LTD depending on their precise timing. The figure shows that synapses with higher release probabilities undergo smaller relative changes in expected strength (Pearson Corr. $r = -0.4416$, $p<0.01$). This suggests that synapses with high release probability are more stable than synapses with low release probability, matching our learning rule.

**Appendix 1—table 1.** Lifelong learning comparison on additional datasets.

Average test accuracies (higher is better, average over all sequentially presented tasks) and standard errors for three repetitions of each experiment on four different lifelong learning tasks for the Presynaptic Consolidation mechanism, Bayesian Gradient Descent (BGD) (*Zeno et al., 2018*) and EWC (*Kirkpatrick et al., 2017*). For the control 'Joint Training', the network is trained on all tasks simultaneously serving as an upper bound of practically achievable performance.

| | Split MNIST | Split fashion | Perm. MNIST | Perm. fashion |
|---|---|---|---|---|
| Presynaptic Consolidation | $82.90^{\pm 0.01}$ | $91.98^{\pm 0.12}$ | $86.14^{\pm 0.67}$ | $75.92^{\pm 0.37}$ |
| No Consolidation | $77.68^{\pm 0.31}$ | $88.76^{\pm 0.45}$ | $79.60^{\pm 0.43}$ | $72.13^{\pm 0.75}$ |
| Bayesian Gradient Descent | $80.44^{\pm 0.45}$ | $89.54^{\pm 0.88}$ | $89.73^{\pm 0.52}$ | $78.45^{\pm 0.15}$ |
| Elastic Weight Consolidation | $70.41^{\pm 4.20}$ | $76.89^{\pm 1.05}$ | $89.58^{\pm 0.53}$ | $77.44^{\pm 0.41}$ |
| Joint Training | $98.55^{\pm 0.10}$ | $97.67^{\pm 0.09}$ | $97.33^{\pm 0.08}$ | $87.33^{\pm 0.07}$ |

