## [Decision Letter]

**Acceptance summary:**

In many nervous systems such as mammalian cortex excitatory synapses are stochastic and the probability of release of neurotransmitter can be modulated by plasticity and neural activity. This paper presents a simple biologically plausible mechanism that regulates the probability of release during learning. Using network simulations the authors show that this can result in more energy efficient processing of learned stimuli by enhancing the reliability of important connections, with lower expected rates of transmission at less important synapses.

**Decision letter after peer review:**

Thank you for submitting your article "Presynaptic Stochasticity Improves Energy Efficiency and Alleviates the Stability-Plasticity Dilemma" for consideration by *eLife*. Your article has been reviewed by 2 peer reviewers, and the evaluation has been overseen by a Reviewing Editor and Timothy Behrens as the Senior Editor. The following individual involved in review of your submission has agreed to reveal their identity: Jean-Pascal Pfister (Reviewer #3).

Essential revisions:

1) Plasticity in p(release) is a nice idea whose simplicity suggests that it is biologically plausible. However, the clear gap in the study is relating the proposed model to the wealth of data on how LTP/D mechanism affect release probabilities. The authors should do more work to survey these mechanisms and, if possible, show how they might relate to the proposed plasticity mechanism or something similar.

2) The generic nature of the networks and simulations is fine but it would be nice in the discussion to suggest what kinds of networks one might expect such a mechanism to be at work.

3) The claim that this mechanism fully alleviates the stability/plasticity dilemma should be tempered: it can impact the tradeoff in specific cases and under broad assumptions may improve energy efficiency. The text should be revised appropriately.

4) Please include a response to the reviewers' comments below with the resubmission and address the specific points raised.

*Reviewer #3 (Recommendations for the authors):*

1. There are several freely available datasets on synaptic plasticity (the ones from Sjostrom is just one example). I would encourage the authors to test their novel learning rule on existing plasticity data sets.

2. I am surprised that the nice work of Fusi is not mentioned here. The cascade model of Fusi precisely proposes a multi-state model for the synapse in order to alleviate the stability-plasticity dilemma. From this perspective, the model proposed by the authors could be seen as a special case of the cascade model.

3. On Figure 2b, I would add an additional control. What happens if the lesioning targets first the lowest expected synaptic strength? Will it be closer to the blue line (Random Order) or the pink line (lowest probability first)?

---

## [Author Response]

Essential revisions:1) Plasticity in p(release) is a nice idea whose simplicity suggests that it is biologically plausible. However, the clear gap in the study is relating the proposed model to the wealth of data on how LTP/D mechanism affect release probabilities. The authors should do more work to survey these mechanisms and, if possible, show how they might relate to the proposed plasticity mechanism or something similar.

We made two changes to relate the model to biological data on presynaptic LTP/D mechanisms. Firstly, in the discussion, we more thoroughly review how individual components of our model relate to known plasticity mechanisms, see subsection “Presynaptic Long-Term Plasticity”. Secondly, we analyse a synaptic plasticity dataset by Sjöström et al., showing some evidence for our stabilisation mechanism, see subsection “Link between Presynaptic Release and Stability”, where we also discuss additional qualitative evidence.

2) The generic nature of the networks and simulations is fine but it would be nice in the discussion to suggest what kinds of networks one might expect such a mechanism to be at work.

We expanded the discussion, explaining where and why the proposed mechanisms could contribute to both energy-efficient processing and lifelong learning, see subsection “Correspondence to Biological Networks”.

3) The claim that this mechanism fully alleviates the stability/plasticity dilemma should be tempered: it can impact the tradeoff in specific cases and under broad assumptions may improve energy efficiency. The text should be revised appropriately.

We updated parts of the title and abstract and went over the remainder of the manuscript to make sure that the claims are tempered.

4) Please include a response to the reviewers' comments below with the resubmission and address the specific points raised.

See detailed comments below.

Reviewer #3 (Recommendations for the authors):1. There are several freely available datasets on synaptic plasticity (the ones from Sjostrom is just one example). I would encourage the authors to test their novel learning rule on existing plasticity data sets.

We added an analysis of a synaptic plasticity dataset by Sjöström et al., (2001), see discussion subsection “Link between Presynaptic Release and Stability” and Figure 9. The analysis is consistent with a key property of our model, namely that synapses with high release probability are more stable than ones with low release probability.

2. I am surprised that the nice work of Fusi is not mentioned here. The cascade model of Fusi precisely proposes a multi-state model for the synapse in order to alleviate the stability-plasticity dilemma. From this perspective, the model proposed by the authors could be seen as a special case of the cascade model.

Thanks for pointing out this oversight, we agree that this is relevant work. We updated the discussion accordingly, see subsection “Related Synapse Models”.

3. On Figure 2b, I would add an additional control. What happens if the lesioning targets first the lowest expected synaptic strength? Will it be closer to the blue line (Random Order) or the pink line (lowest probability first)?

We added this control. Keeping “high energy (expected strength)” synapses is even more important for the network than keeping “high release probability synapses”. Nevertheless, the claim that release probabilities are related to importance holds true, and our analysis also shows that release probabilities are more closely related to the Fisher information (which is a more local measure than the lesion experiment) than to the expected strength.